# Clinical Implications of the Genetic Background in Pediatric Pulmonary Arterial Hypertension: Data from the Spanish REHIPED Registry

**DOI:** 10.3390/ijms231810433

**Published:** 2022-09-09

**Authors:** Alejandro Cruz-Utrilla, Natalia Gallego-Zazo, Jair Antonio Tenorio-Castaño, Inmaculada Guillén, Alba Torrent-Vernetta, Amparo Moya-Bonora, Carlos Labrandero, María Elvira Garrido-Lestache Rodríguez-Monte, Alejandro Rodríguez-Ogando, María del Mar Rodríguez Vázquez Del Rey, Juana Espín, Beatriz Plata-Izquierdo, María Álvarez-Fuente, Antonio Moreno-Galdó, Pilar Escribano-Subias, María Jesús Del Cerro Marín

**Affiliations:** 1Pulmonary Hypertension Unit, ERN-Lung, Cardiology Department, Hospital Universitario 12 de Octubre, 28041 Madrid, Spain; 2Instituto de Genética Médica y Molecular (INGEMM), Hospital Universitario La Paz, 28046 Madrid, Spain; 3CIBERER, Centro de Investigación en Red de Enfermedades Raras, Instituto de Salud Carlos III, 28029 Madrid, Spain; 4ITHACA, European Reference Network on Rare Congenital Malformations and Rare Intellectual Disability, 1000 Brussels, Belgium; 5Pediatric Cardiology Unit, Department of Pediatrics, Hospital Universitario Virgen del Rocío, 41013 Sevilla, Spain; 6Pediatric Allergy and Pulmonology Section, Department of Pediatrics, Vall d’Hebron Hospital Universitari, Vall d’Hebron Barcelona Hospital Campus, Universitat Autònoma de Barcelona, 08035 Barcelona, Spain; 7Pediatric Cardiology, Department of Pediatrics, Hospital Universitari i Politècnic La Fe, 46026 Valencia, Spain; 8Pediatric Cardiology, Department of Pediatrics, Hospital Universitario La Paz, 28046 Madrid, Spain; 9Pediatric Cardiology and Adult Congenital Heart Disease Department, Ramón y Cajal University Hospital, Instituto Ramón y Cajal de Investigación Sanitaria (IRYCIS), 28034 Madrid, Spain; 10Pediatric Cardiology, Department of Pediatrics, Hospital Universitario Gregorio Marañón, 28009 Madrid, Spain; 11Pediatric Cardiology, Hospital Universitario Virgen de las Nieves, 18014 Granada, Spain; 12Pediatric Cardiology, Hospital Universitario Virgen de la Arrixaca, 30120 Murcia, Spain; 13Pediatric Cardiology, Hospital Universitario de Salamanca, Instituto de Investigación Biomédica de Salamanca (IBSAL), Facultad de Medicina, Universidad de Salamanca, 37007 Salamanca, Spain; 14Centro de Investigación Biomédica en Red de Enfermedades Cardiovasculares (CIBERCV), 28029 Madrid, Spain

**Keywords:** pediatric pulmonary hypertension, genetics, heritable pulmonary arterial hypertension, pulmonary veno-occlusive disease

## Abstract

Background: Pulmonary arterial hypertension (PAH) is a severe and rare disease with an important genetic background. The influence of genetic testing in the clinical classification of pediatric PAH is not well known and genetics could influence management and prognosis. Objectives: The aim of this work was to identify the molecular fingerprint of PH children in the *REgistro de pacientes con HIpertensión Pulmonar PEDiátrica* (REHIPED), and to investigate if genetics could have an impact in clinical reclassification and prognosis. Methods: We included pediatric patients with a genetic analysis from REHIPED. From 2011 onward, successive genetic techniques have been carried out. Before genetic diagnosis, patients were classified according to their clinical and hemodynamic data in five groups. After genetic analysis, the patients were reclassified. The impact of genetics in survival free of lung transplantation was estimated by Kaplan–Meier curves. Results: Ninety-eight patients were included for the analysis. Before the genetic diagnoses, there were idiopathic PAH forms in 53.1%, PAH associated with congenital heart disease in 30.6%, pulmonary veno-occlusive disease—PVOD—in 6.1%, familial PAH in 5.1%, and associated forms with multisystemic disorders—MSD—in 5.1% of the patients. Pathogenic or likely pathogenic variants were found in 44 patients (44.9%). After a genetic analysis, 28.6% of the cohort was “reclassified”, with the groups of heritable PAH, heritable PVOD, TBX4, and MSD increasing up to 18.4%, 8.2%, 4.1%, and 12.2%, respectively. The MSD forms had the worst survival rates, followed by PVOD. Conclusions: Genetic testing changed the clinical classification of a significant proportion of patients. This reclassification showed relevant prognostic implications.

## 1. Introduction

Pulmonary arterial hypertension (PAH) is a severe and rare disease, still characterized by high mortality despite important therapeutic advances [1]. Endothelial dysfunction, cell proliferation, and inflammation are some of the mechanisms involved in this heterogeneous condition [2]. The current classifications of pulmonary hypertension (PH) are based mainly on the clinical associated conditions [3]. In pediatric patients, PH reveals a different disease distribution, where heritable, as well as associated cases with congenital heart disease (CHD) or development lung disorders (DLD), are overrepresented in comparison with adults [4,5,6,7]. Equally, the genetic architecture is singular and probably much more relevant in pediatric-onset PAH, considering that the number of genetic variants is extraordinarily higher in pediatric patients [8,9].

During the recent years, genetics has been gaining importance in the complex pathobiology of this entity. Multiple molecular pathways have been related to the development of PAH, considering the discovery of genes comprising different signaling pathways. Although the most frequently affected molecular pathway in this disease is the transforming growth factor-ß (TGFß) superfamily, other relevant pathways have been identified [10]. Importantly, the investigation of the role of genetics in PAH has guided the identification of novel therapies, demonstrated by the potential utility of sotatercept for PAH acting on the TGFß pathway [11]. Nevertheless, the current classifications are based fundamentally on the clinical phenotype, neglecting the genetic influence in PAH [12]. Additionally, there is still an unmet need for dedicated studies evaluating the influence of genetics in children despite age differences. Here, we highlight the relevance of the genetic background on the clinical phenotype of pediatric PH, as well as its possible impact on the reclassification of patients, which could further influence the management and prognosis in this rare condition.

## 2. Methods

### 2.1. Study Population

The Spanish Registry for Pediatric Pulmonary Hypertension (REHIPED) includes patients since January 2009. Patients up to 18 years at diagnosis from January 2011 to December 2021 were included in this study. The diagnosis of PAH required a right heart catheterization (RHC) with a mean pulmonary artery pressure ≥ 25 mmHg, pulmonary vascular resistance index ≥ 3 WU·m^2^, and a pulmonary artery wedge pressure ≤ 15 mmHg. In cases in which the RHC was not performed (e.g., an unstable patient or Eisenmenger physiology), the patient was included by means of a confirmatory transthoracic echocardiography or pathology sample, as well as clinical recordings, evaluated by the registry committee. Parental written informed consent was obtained in all cases. The protocol was approved by institutional review boards and ethics committees in all the participant centers^7^. Clinical variables were collected from the REHIPED registry.

### 2.2. Clinical Classification of Included Patients

Before a genetic diagnosis, the patients were classified into five groups: idiopathic PAH (IPAH), heritable PAH (HPAH), pulmonary veno-occlusive disease (PVOD), PAH associated with CHD (PAH-CHD), and other forms including multisystemic disorders (MSD) or DLD. Cases with a known familial aggregation of the disease were classified as HPAH. PAH-CHD was categorized into four groups [13].

After the genetic study, the patients were reclassified according to the classification proposed by the latest World Symposium on Pulmonary Hypertension, with some exceptions. Patients with pathogenic (P) or likely pathogenic (LP) variants in the *TBX4* gene were considered a separate group, considering they could have differential features and might be classified as “heritable” PAH, PAH associated to DLD or even as PAH associated with CHD. Patients with genetic variants in the *GBE1*, *FOXF1*, *MECP2*, *MMACHC*, *NFU1*, and *VHL* genes were considered as having a PH form associated with MSD. Patients with genetic variants in genes related to PAH were reclassified into the HPAH form. PVOD was confirmed in all cases by biallelic variants in the *EIF2AK4* gene.

### 2.3. Genetic Analyses

From 2011 onward, Sanger sequencing and multiplex ligation-dependent probe amplification were used to detect genetic variants in the *BMPR2*, *TBX4*, and *KCNK3* genes. Since 2014, a customized next generation sequencing (NGS) panel of 21 genes (HAP v1.2) was applied [14]. In 2017 this panel was expanded to cover 35 genes (HAP v2), and finally in 2019, we started with HAP v3, including 37 genes. Since 2020, whole-exome sequencing has been applied for the previously followed-up patients without known P/LP genetic variants after prior genetic analyses, and for each incident case (Figure 1).

Classification of the genetic variants was completed following the American College of Medical Genetics and Genomics guidelines [15]. Genetic counseling was provided for each included patient and their related familiars. A family history was obtained during the first genetic study. A blood sample for analysis was obtained only in probands. When a genetic variant was discovered, the study of the first-degree relatives was offered to analyze the possibility of co-segregation.

### 2.4. Outcomes

Follow-up data is recorded in the REHIPED registry every 6 months. The main outcome was the first occurrence of a lung transplantation or death.

### 2.5. Statistics

The categorical variables are reported as absolute and relative frequencies, and compared with Pearson’s or Fisher’s exact tests, as appropriate. The continuous variables are reported as means (standard deviation) or medians (interquartile range) and compared with t-tests or Mann–Whitney U tests, as appropriate. Kaplan–Meier survival curves were compared using the log-rank test. Stata version 14.0 (StataCorp, College Station, TX, USA) and R studio (v 4.0.3) were used for the data analysis.

## 3. Results

Between January 2011 and December 2021, 285 patients were included in the REHIPED registry as PAH or PH related with MSD. One-hundred-and-six individuals up to 18 years at diagnosis were initially genetically studied and included from the REHIPED cohort. After excluding eight non-valid patients, ninety-eight patients were selected (Figure 2).

In the selected cohort of 98 patients (56.1% female), the median age at diagnosis was 7.1 years (IQR 1.5–14.7), and Caucasian was the predominant population (81.6%). Before the genetic testing, the patients were classified as idiopathic in 53.1%, associated with CHD in 30.6%, as heritable PVOD in 6.1%, familial PAH in 5.1%, and MSD associated with PH in 5.1% of cases. There were global differences in the descendance between the PH groups. Caucasian was the most frequent descendance in all PH groups except in PVOD, in which the Romani ethnicity was predominant. Regarding the mode of diagnosis, most of the patients were diagnosed as an index case, and HPAH and PVOD were more frequently diagnosed after family screening. Haemodynamically, there were no differences between the variables analyzed except for the cardiac index, which was slightly lower in patients with PVOD, and PH in MSD, when compared with the other PH groups (Table 1).

We found pathogenic or likely pathogenic variants in 44 of the screened patients (44.9%). The genetic findings in each patient are described extensively in Appendix A. After genetic analysis, 28 patients (28.6% of the cohort) were “reclassified” according to the pediatric classification of PH, with the groups of HPAH, heritable PVOD, TBX4, and PH in MSD increasing up to 18.4%, 8.2%, 4.1%, and 12.2%, respectively (Figure 3, and Table 1). Interestingly, the age at PH diagnosis was significantly different depending on the classification after genetic testing. While those patients diagnosed with a *FOXF1*, *MECP2*, *NFU1*, or other MSD, as well as *TBX4* variants, were, in their vast majority, newborns or infants under 2 years at PH diagnosis, the age at diagnosis was generally higher in the other PAH types.

### 3.1. Genetic Findings in Idiopathic, Heritable PAH and TBX4 Forms

Among the 28 included children with a final diagnosis of IPAH, there were eight genetic variants in 7 patients. Seven out of eight were classified as variants of unknown significance—VUS—and one child had an *EIF2AK4* heterozygous pathogenic variant (Appendix A).

The group of HPAH patients after genetic studies included eighteen individuals. Four of the eighteen individuals were diagnosed after family screening in known familial PH forms (22.2% of cases in this group): three of them had a P/LP genetic variant (two variants in *BMPR2* and one variant in *AQP1*) [16], and one of them did not show a genetic variant. Additionally, fourteen had sporadic PAH forms, being classified as HPAH after discovering a P/LP variant. Globally, *BMPR2* was the most frequently affected gene (10 cases, 64.7%). Two patients had PAH associated with hemorrhagic hereditary telangiectasia (HHT). Of them, one patient had an *ACVRL1* variant, and the other one showed a pathogenic homozygous variant in *GDF2*. Both patients were the index case, were initially classified as IPAH, manifested recurrent epistaxis, and had multiple telangiectasias in uncommon locations, being clinically classified as HHT by Curaçao criteria [17].

Among the four patients in whom a *TBX4* variant was discovered, three of them were initially diagnosed with an IPAH (two cases during the neonatal period and one at 2 months of age). The remaining TBX4 patient was clinically diagnosed with PAH associated with an incidental septal defect+ at the age of 15 years.

### 3.2. Findings in the PAH-CHD Population

Overall, eight variants were found in seven patients of the twenty-eight included individuals in this group (25.0%). Of them, three had at least one P or LP variant: one patient with PAH after the closure of a ventricular septal defect (*SOX17*), one individual with PAH associated with incidental defects (*ABCC8* and *SMAD1*), and the remaining one with a complex Eisenmenger syndrome (*BMPR2*).

In this group, four patients had a trisomy in chromosome 21, and two additional patients had other types of chromosomic disorders (e.g., 7p22.1 duplication and a 4q35 duplication). The description of the variants found in each patient with PAH-CHD is shown in Appendix A.

### 3.3. Genetic Findings in the Pediatric PVOD Population

PVOD was initially suspected in six patients and confirmed in five. Three of them were diagnosed after the genetic screening of the first-degree relatives of adults with PVOD. Additionally, the disease was suspected in one adolescent due to the presence of PH, chronic respiratory insufficiency, probable findings of PVOD in the computed tomography, and a severe impairment of the diffusing capacity of the lung for carbon monoxide (DLCO) of 30%, being reclassified as a Cobalamin C deficiency (*MMACHC* variant) after the genetic testing. Interestingly, the previous findings improved considerably in this patient after specific supplementation for this rare condition (Appendix A).

In total, eight children were identified as a heritable PVOD during the study period. The disease was confirmed by the presence of homozygous variants in all of them after genetic analysis. The most frequent genetic variant was the founder mutation (Pro115Leu) in the *EIF2AK4* gene (75.0%), discovered in all these cases in patients of the Romani ethnicity. Another two different biallelic variants were found in Caucasian patients: a homozygous pathogenic variant (p.Lys187fsTer9), and a compound heterozygous variant involving p.Arg585Ter and p.Pro115Leu. In this last patient, the offspring of a non-consanguineous couple, PVOD was confirmed during the study of lung samples after bilateral lung transplantation.

### 3.4. Genetic Analysis in Patients with Pulmonary Hypertension Associated with a Multisystemic Disorder

This group included twelve patients: four *NFU1* variants, three variants in *MECP2*, two patients with *FOXF1* variants, one individual with an *MMACHC* variant, another one with a *GBE1* variant, and one patient with Von Hippel-Lindau disease (VHL). In most of them, the diagnosis of PH was made during the first 6 months of life.

Seven patients of the twelve were diagnosed with PH before the MSD. Five of these cases initially mimicked an IPAH form of a neonatal debut, and finally had a PH form associated with an *NFU1* homozygous variant (four cases) or associated with a *MECP2* variant (one case). An additional patient was initially diagnosed with PAH associated with CHD (partial auriculoventricular canal), being reclassified as a PH associated with alveolar capillary dysplasia with a “misalignment of veins” after autopsy and the result of the genetic analysis.

In the remaining five patients (two cases of MECP2 syndrome, one patient with alveolar capillary dysplasia with misalignment of veins, one glycogen storage disease, and one patient with VHL), PH was found after the discovery of associated multisystemic findings (Appendix A).

### 3.5. Differences in Survival Free of Death or Lung Transplantation according to the Clinical Classification and Genetics after Genetic Testing

Analyzing the differences in survival according to the final clinical diagnosis after the genetic study, PH-MSD had the worst survival (median survival of 0.12 years, 0.07–1.48. Figure 4A). When we further analyzed the survival in those patients and according to the specific genetic variant found, *FOXF1* and *NFU1* had a worse long-term survival in comparison with the *MECP2* variants (median survival of 0.09 years, 0.07–0.11 for *NFU1* and *FOXF1* in comparison with 1.45 years, 0.87–1.52; *p* = 0.020 for the comparison between *FOXF1*-*NFU1* combined and *MECP2*. Figure 4B).

The PVOD patients also had a significantly worse survival free of lung transplantation when compared with the patients with heritable or IPAH (median survival of 1.26 years, 0.52–1.96; *p* < 0.001 for the comparison with IPAH or HPAH, Figure 4A). Patients with variants in *TBX4* showed an excellent long-term survival (5.35 years, 1.81–17.27; Figure 4B).

## 4. Discussion

This study analyzes the evolving genetic landscape in a large multicenter cohort of pediatric PAH patients over twelve years, showing interesting differences in the genetic background of the Spanish pediatric population in comparison with previous cohorts (Table 2) [6,9,18,19,20].

The diagnostic yield of genetic analyses in our study was relatively high despite the inclusion of a significant number of cases of PAH-CHD, a group in which the genetic background had not been completely explored yet. Globally, we found a high number of variants as a result of the introduction of new techniques over time, enhancing the need for searching for new molecular pathways involved in this disease. In IPAH and HPAH, the *BMPR2* gene was the most frequent mutation, in concordance with other published registries. Nevertheless, the Spanish population showed a lower prevalence of *TBX4* compared with the Dutch registry, and a lower frequency of the *ACVRL1* gene (*ALK1*) than the Japanese pediatric PAH population [6,18,20]. These different results could be related to a different genetic basis in the pediatric Iberian population. In contrast with other registries, pediatric PVOD in our series was highly represented (9.1% of the total cohort). A vast majority of the PVOD patients belonged to the Romani ethnic group (six out of eight), a highly inbred population. Interestingly, up to five cases (62.5%) of them were the index case, diagnosed after genetic testing despite an initial diagnosis of idiopathic or familial PAH. Genetically, those patients were characterized by the presence of the homozygous founder mutation in *EIF2AK4* in six cases, representing the offspring of the adult Spanish PVOD population [21]. Nevertheless, two other genetic findings in *EIF2AK4* were described in patients of Caucasic origin: a heterozygous compound patient and a homozygous variant, results that increase the need of maintaining a high clinical suspicion for PVOD even in patients without a previous family history. Difficult to diagnose in adults [22], this entity is even more difficult to diagnose during childhood, as the implementation of DLCO testing is difficult in infants or small children. Additionally, other conditions in pediatrics could simulate PVOD, such as the described case of Cyanocobalamin C deficiency in this work, or NFU mitochondrial disease. Consequently, performing genetic testing on each infant or child with suspected PAH is crucial to allow for proper diagnosis and treatment of this entity. The percentage of patients with heritable PAH or PVOD in whom the diagnosis was made after the family screening, was relatively small in our cohort (22.2% and 37.5%, respectively). Nonetheless, the age at diagnosis (10.2 and 15.7 years, respectively) reinforces the need for the screening of asymptomatic carriers of pathogenic variants even during early childhood. Noteworthy, an incidence of PAH of 2.3% per year was reported in asymptomatic carriers of variants in *BMPR2* after dedicated follow-up in the French PAH network [23].

This study is the largest evaluating genetics in PAH-CHD, including a heterogeneous cohort of cases among the four subtypes of PAH-CHD. We found genetic variants in 25% of this group. We found a digenic likely pathogenic variant (*ABCC8/ SMAD 1*) in a case of coincidental PAH and several thoracic collaterals, a pathogenic variant in *BMPR2* in a case of PAH after defect closure, and a pathogenic *SOX17* variant in a patient with Eisenmenger syndrome. Discovering a *BMPR2* variant in PAH after defect closure reinforces the theses defending the use of genetics before considering the defect correction, as pathogenic variants in this gene have been associated with the development of pulmonary vascular disease in CHD [24,25]. Equally, the development of severe pulmonary vascular remodeling conducted to Eisenmenger physiology at a young age in a patient with a pathogenic variant in *SOX17* should prompt the inclusion of genetic testing in the evaluation of CHD and disproportionate PH, as genetic mechanisms could be part of the complex pathogenesis of this entity [26]. The presence of significant thoracic collaterals and related hemoptysis could also be a marker of a significant genetic background, probably related to a long clinical history and a higher hemodynamic severity [27].

Another important finding was the high frequency of MSD or DLD found associated with PH, especially in the subgroup of infants under 2 years. In 58.3% of them, PH was the main manifestation, mimicking IPAH. The inclusion of genes related to these disorders could facilitate a prompt diagnosis, with important treatment and prognostic implications, since most of these patients had an early clinical debut and extremely high mortality. In the first published report on the outcomes of pediatric PH in Spain [7], we described higher mortality in PH patients under 2 years, so the prevalence of MSD-related genes could explain these outcomes. The proper identification of some of these genes could allow us to use specific treatments that potentially improve prognosis (as in the case of Cobalamin C deficiency), result in a redirection of care (*NFU* mitochondrial disease), or even in the consideration of an early Potts shunt or lung transplantation (in cases of *FOXF1* variants). Mitochondrial diseases, inborn errors of metabolism, neurodevelopmental disorders, and glycogen storage diseases (GSD), such as those caused by *NFU1*, Cobalamin C (CblC) disease, *MECP2* variants, or *GBE1*, respectively, should be covered by the genetic techniques used in childhood-onset cases of PAH [28,29,30,31,32]. Of relevance, although PH has been described in multiple cases of GSD type 1 or type 2 [33,34], as far as we know we describe here for the first time the association of PH in a newborn with a genetic variant in *GBE1*, diagnostic of a GSD type 4. This patient died during his first year of life, had a severe neonatal hypotonia, and pathologically was characterized by the predominantly venous involvement (PVOD). Regarding the *MECP2* variants, duplication syndrome has been described among males with PH during early childhood [35]. Additionally, De Felice et al. described some years ago the presence of multiple types of lung lesions in patients with classic Rett Syndrome [36]. We describe here two females with loss of function variants in this gene, diagnosed during their first month of life. Interestingly, one of these cases was also pathologically analyzed, presenting extensive arteriolar and venous involvement. These two cases add interesting new data about pulmonary vascular disease across the whole *MECP2* genetic spectrum. We found a case of PH in a patient with VHL and a homozygous variant in *VHL*. PH in this disease has been described in rare cases, both in patients with homozygous and compound heterozygous variants [37,38].

The genetic background of each patient often determines the assigned phenotype. After the genetic study, a significant proportion of patients were moved into a different clinical category. Overlapping is one of the main problems of the current clinical classifications of PH. Patients with variants in *TBX4* are often diagnosed with persistent pulmonary hypertension of the newborn (PPHN), but when diagnosed in late childhood or adulthood, they are usually diagnosed as IPAH. Even these cases could have associated coincidental septal defects or could have an abnormal DLCO or lung abnormalities in their CT scan, being classified as PAH-CHD, PVOD, or PH related to interstitial lung disease [39,40]. Patients carrying pathogenic variants in genes associated with MSD showed a younger age at diagnosis in comparison with the rest of the groups. Additionally, most of the TBX4 patients had a younger age at presentation compared with IPAH, HPAH, or PAH-CHD. On the contrary, the PVOD patients showed a median age at diagnosis above fifteen years. This different age of presentation depending on the gene was also demonstrated by Zhu et al. [9]. The genetic background should always be studied and incorporated into clinical classifications to improve our knowledge of the PH spectrum and guide the management of it. Considering the age paradigm, PVOD should be discarded in adolescent patients, and TBX4 and MSD must be ruled out in newborns and infants with severe PH.

In pediatric cases with pulmonary hypertension, the ideal approach is to collect blood from the index case and their parents. Genetic panels to study PH in pediatrics should include not only genes with high evidence of association with PAH, but also genes associated with MSD, especially in neonates and infants, considering that PH could be the first or the predominant manifestation of these disorders. When no variant is detected in these genes, whole exome sequencing (WES) or whole genome sequencing (WGS), including parent’s material to detect de novo variants, should be performed, contributing not only to the diagnosis of the index case, but also to find new genes potentially involved in the development of the disease [41].

### Limitations

The study could be depicting a clinical and genetical image of severe cases instead of a global picture of genetics in pediatric PAH, as mild cases of PH could go unnoticed; however, the vast majority of forms of PH clinically detected in Spain are included in the REHIPED registry, and genetic studies were performed in approximately one-fifth of the total number of patients in the registry. PVOD was mostly found in Romani patients; therefore, the extrapolation can be difficult to other environments. Our study cannot statistically prove that the genetic testing improved outcomes, as it was not designed with this purpose. The study has additional limitations inherent to its observational nature.

## 5. Conclusions

The genetic background of the Spanish PAH pediatric population confirmed BMPR2 as the most common cause of heritable PAH, showing differential features in comparison with previously reported studies in other countries: a higher prevalence of PVOD due to biallelic variants in *EIF2AK4*, and a lower prevalence of variants in *TBX4*. We found a significant number of variants in the PAH-CHD patients, across the entire spectrum of cases associated with CHD. The incorporation of genes related to multisystemic, and developmental lung disorders and novel molecular techniques allowed an accurate and early diagnosis of several multisystemic disorders mimicking IPAH. Genetic testing changed the clinical classification of a significant proportion of patients, with prognostic implications. Further development of genetic testing, as well as collaborative efforts between research groups, will enhance the use of genetics in the management of patients with different types of pulmonary hypertension. Consequently, the genetic background should be incorporated in future classifications of pediatric pulmonary hypertension.

## Figures and Tables

**Figure 1 ijms-23-10433-f001:**
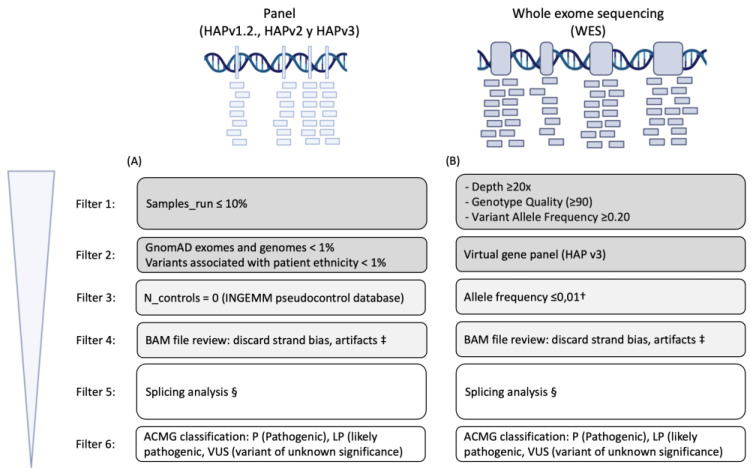
**Workflow for variant prioritization. Schematic representation of the pipeline for tertiary/prioritization analysis of panel (A) and WES (B) samples**. Custom NGS panels (HAPv1.2, NGS v2, and NGS v3) were designed with NimbleDesign (Roche, Indianapolis, IN, USA). Fragmentation and library preparation was performed with a SeqCap EZ Choice Enrichment Kit (Roche, Indianapolis, IN, USA), and sequencing was performed with the Illumina MiSeq platform (Illumina, San Diego, CA, USA). Genes included in HAPv1.2.: ACVRL1, GDF2, BMPR1B, BMPR2, CAV1, EIF2AK4, ENG, KCNA5, KCNK3, NOTCH3, SMAD1, SMAD4, SMAD5, SMAD9, TBX4, TOPBP1, SARS2, CPS1, ABCC8, CBLN2, and MMACHC; HAPv2: those included in HAPv1.2. and KLF2, NOTCH1, FOXF1, COX5A, CCDC80, HRG, VCAN, NRG1, CTCFL, APOH, MYSM1, FCER2, and CYP1A1; HAPv3: those included in HAPv2 and AQP1, ATP13A3 and SOX17. Library preparation for whole exome sequencing was carried out by Agilent SureSelect TM (v 6.0), and all exon kits were followed by sequencing in a NovaSeq Sequencer (Illumina, USA). The exomes were analyzed by VarSeq (Golden Helix, Bozeman, MT, USA) to detect both single nucleotide variants (SNVs) and copy number variants. A three-step prioritization algorithm was performed. † Only variants who have had rare allele frequency (≤0.01) in control population databases such as gnomAD exomes (v2.1.1), gnomAD genomes (v2.1.1), Kaviar (version 160204-Public), Bravo, were kept. ‡ BAM review was performed through Alamut and IGV; § Splicing analysis through SpliceSiteFinder-like, MaxEntScan, GeneSplicer, NNSPLICE.

**Figure 2 ijms-23-10433-f002:**
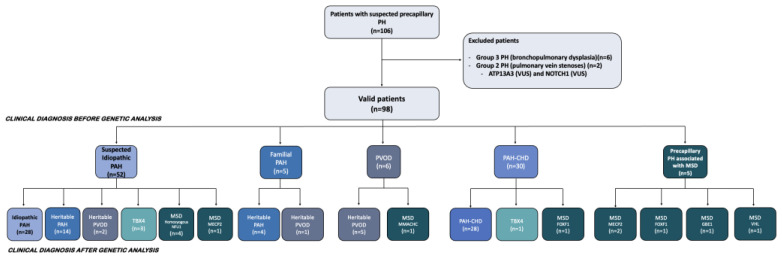
**Flow-chart of included patients and the definitive diagnosis after genetic testing.** Patients in the REHIPED registry with genetic analysis and classified as a group 2 or group 3 pulmonary hypertension were excluded. FOXF1 (Forkhead Box F1 gene), GBE1 (1,4-Alpha-Glucan Branching Enzyme 1 gene), MECP2 (Methyl-CpG Binding Protein 2 gene), MMACHC (Methylmalonic aciduria and homocystinuria type C protein), MSD (multisystemic disorders), PAH (pulmonary arterial hypertension), PAH-CHD (pulmonary arterial hypertension associated with congenital heart disease), PH (pulmonary hypertension), PVOD (pulmonary veno-occlusive disease), TBX4 (T-Box Transcription Factor 4), VUS (variant of unknown significance), VHL (Von-Hippel-Lindau gene).

**Figure 3 ijms-23-10433-f003:**
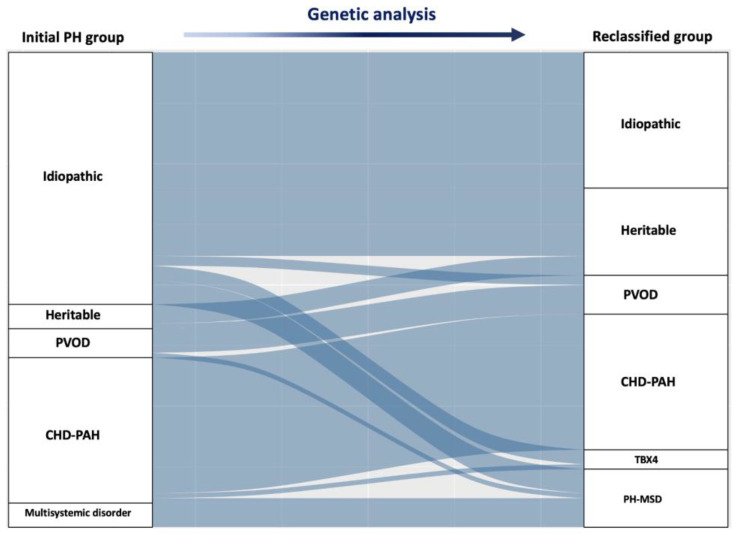
**Sankey diagram demonstrating the reclassification of patients after genetic analysis.** The left column shows the clinical classification prior to genetic analysis. The right column reflects the clinical classification after reclassification. PAH (pulmonary arterial hypertension); PAH-CHD (pulmonary arterial hypertension associated with congenital heart disease); PVOD (pulmonary veno-occlusive disease). The size of each section represents the number of subjects in each classification; for example, the number of subjects classified with IPAH was lower after reclassification according to genetic testing results.

**Figure 4 ijms-23-10433-f004:**
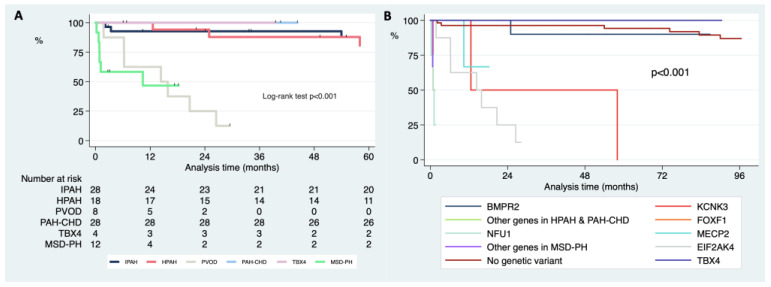
**Kaplan–Meier curves of free survival of death or lung transplantation comparing the different types of pediatric PAH after genetic testing (A) and comparing exclusively the specific genetic background (B).** DLD (development lung disorders), PAH (pulmonary arterial hypertension); PAH-CHD (pulmonary arterial hypertension associated with congenital heart disease); PVOD (pulmonary veno-occlusive disease).

**Table 1 ijms-23-10433-t001:** **Clinical, demographic and hemodynamical characteristics of the studied population.** The clinical PH group shown in this table refers to the classification after genetic study. CI (cardiac index); DLD (developmental lung disease); IQr (interquartile range); iPVR (index pulmonary vascular resistance); iSVR (index systemic vascular resistance); mPAP (mean pulmonary artery pressure); N/A (not available); RAP (right atrial pressure); WU (Wood units). * *p*-value refers to global differences between groups.

	Total Cohort (n = 98)	Idiopathic (n = 28)	Heritable PAH (n = 18)	PVOD (n = 8)	PAH-CHD (n = 28)	TBX4 (n = 4)	PH-MSD (n = 12)	*p* Value
**Age—years (median—IQr)**	7.1 (1.5–14.7)	6.3 (3.6–12.6)	10.2 (5.6–17.7)	15.7 (12.4–17.3)	6.3 (2.1–10.5)	0.4 (0.2–8.1)	0.1 (0.1–0.4)	<0.001
**Gender—male (n—%)**	43 (43.9)	13 (46.4)	6 (33.3)	6 (75.0)	10 (35.7)	2 (50.0)	6 (50.0)	0.412
**Descendance** **(n—%)**								<0.001 *
**Arabic**	2 (2.1)	0 (0.0)	0 (0.0)	0 (0.0)	0 (0.0)	0 (0.0)	2 (16.7)	
**Asiatic**	1 (1.0)	0 (0.0)	1 (5.6)	0 (0.0)	0 (0.0)	0 (0.0)	0 (0.0)	
**Caucasian**	80 (81.6)	27 (96.4)	13 (72.2)	2 (25.0)	25 (89.3)	4 (100.0)	9 (75.0)	
**Romani**	8 (8.2)	0 (0.0)	2 (11.1)	6 (75.0)	0 (0.0)	0 (0.0)	0 (0.0)	
**Hispanic**	6 (6.1)	1 (3.6)	2 (11.1)	0 (0.0)	2 (7.1)	0 (0.0)	1 (8.3)	
**Black**	1 (1.0)	0 (0.0)	0 (0.0)	0 (0.0)	1 (3.6)	0 (0.0)	0 (0.0)	
**Mode of diagnosis—Index case vs. family screening (n—%)**	91 (92.9)	28 (100.0)	14 (77.8)	5 (62.5)	28 (100.0)	4 (100.0)	12 (100.0)	0.001 *
**Baseline**								
**hemodynamics**	N = 82	N = 27	N = 16	N = 7	N= 21	N = 3	N = 8	
**mPAP—mmHg (median—IQr)**	53.0 (40.0–63.0)	54.0 (39.0–70.0)	55.0 (42.5–68.5)	44.0 (22.0–56.0)	61.0 (46.0–65.0)	39.0 (27.0–61.0)	41.0 (32.0–49.5)	0.081
**RAP—mmHg (median—IQr)**	7.0 (5.0–10.0)	7.0 (6.0–10.0)	5.5 (2.0–11.5)	7.0 (4.0–10.0)	7.5 (6.0–10.0)	11.0 (8.0–12.0)	6.5 (5.0–12.0)	0.814
**iPVR—WU * m2 (median—IQr)**	11.3 (7.0–17.4)	11.7 (7.9–16.9)	15.6 (7.0–20.4)	7.2 (3.2–13.7)	11.8 (5.4–18.3)	10.8 (3.8–16.2)	8.6 (7.1–14.3)	0.500
**CI—L/m/m2**	3.1 (2.0–4.0)	3.1 (1.9–3.9)	2.6 (2.0–3.2)	2.3 (1.8–2.7)	4.0 (3.1–4.8)	6.4 (4.8–8.0)	2.2 (1.6–3.1)	0.007
**(Median—IQr)**								
**iPVR/ iSVR (median—IQr)**	0.9 (0.6–1.0)	0.8 (0.6–0.9)	0.8 (0.6–0.9)	N/A	1.0 (0.6–1.3)	0.8 (0.8–2.2)	0.9 (0.8–1.0)	0.178

**Table 2 ijms-23-10433-t002:** **Comparison between published studies analyzing genetics in pediatric pulmonary hypertension.** CHD (congenital heart disease); FPAH (familial pulmonary arterial hypertension); HPAH (heritable pulmonary arterial hypertension); IPAH (idiopathic pulmonary arterial hypertension); N/A (not available); PH (pulmonary hypertension); PVOD (pulmonary veno-occlusive disease). * Pathogenic or likely pathogenic variants. ** Only includes heritable cases after genetic analysis (includes familial cases without any significant genetic variant and heritable cases with pathogenic or likely pathogenic variants).

	Evaluated Genes	Number of Patients	PH Groups (IPAH, HPAH, PVOD, CHD, Others)	Diagnostic Yield of Genetic Analysis *	Results in IPAH/HPAH	Results in PVOD	Results in CHD-PAH	Age at Diagnosis of Mutation Carriers (Mean)
**Chida et al., 2012** [18]	BMPR2, ACVRL1, SMAD8, BMPR1B	57	46/10/0/0/0	25/54 (46.3%)	BMPR2 n = 13/57 (33%) ALK1 n = 6/57 (13%) SMAD8 n = 1/57 (1.7%) BMPR1B n = 2 (3.5%)	NA	NA	9.2
**Levy et al., 2016** [19]	BMPR2, ACVRL1, TBX4, KCNK3, EIF2AK4	42	35/5/3/23	14/42 (33.3%)	BMPR2 n = 5/40 (12.5%) ACVRL1 n = 4/40 (10%) TBX4 n = 3/40 (7.5%)	EIF2AK4 2/3 (67.7%)	0/23 (0%)	8.3
**Zhu et al., 2018** [9]	BMPR2, TBX4, ACVRL1, BMPR1B, CAV1, EIF2AK4, ENG, KCNK3, SMAD4, SMAD9	155	130/25	48/155 (31.0%)	BMPR2 n = 27/155 (17.7%) TBX4 n = 12/155 (7.4%) ACVRL1 n = 3/155 (1.9%), KCNK3 n = 2/155 (1.3%), SMAD9 n = 1/155 (0.6%)	N/A	N/A	
**Haarman et al., 2020** [20]	BMPR2, ACVRL1, CAV1, ENG, KCNK3, SMAD9, TBX4, EIF2AK4, VHL, MMACHC, CBLC, ACTA2	70	19/16/5/20/10	19/70 (27.1%)	BMPR2 n = 7/35 (20%) TBX4 n = 7/35 (20%) ACVRL1 n = 1/35 (2.8%), PTPN11 n = 2/35 (5.7%) KCNK3 n = 1/35 (2.8%)	EIF2AK4 2/5 (40.0%)	ACTA2 1/20 (5%)	2.8 years TBX4 and14.0 years in BMPR2 carriers
**PPHNet Registry. 2021** [6]	BMPR2, ACVRL1, SMAD9, CAV1, KCNK3, TBX4, GDF2	40 FPAH	0/40/0/0/0	36/40 heritable cases (90.0%) **	BMPR2 n = 17/40 (43%) TBX4 n = 6/40 (15%) ALK/ENG n = 5/40 (5%) GDF2 n = 2 /40 (5%) CAV1 n = 1/40 (3%) KCNK3 n = 1 (3%)	N/A	N/A	N/A
**REHIPED registry (this study)**	ABCC8, ACVRL1, BMPR1B, BMPR2, CAV1, CBLN2, CPS1, EIF2AK4, ENG, GDF2, KCNA5, KCNK3, MMACHC, NOTCH3, SARS2, SMAD1, SMAD4, SMAD5, SMAD9, TBX4, and TOPBP1	98	28/18/8/28/16	44/98 (44.9%)	BMPR2 n = 12/50 (24.0%) TBX4 n = 4/50 (8.0%) KCNK3 n = 2/50 (4.0%) ACVRL1 n = 1 (2.0%) BMPR1B n = 1 (2.0%) GDF2 n = 1 (2.0%)	EIF2AK4N = 8/8 (100.0%)	N = 7/28 (25.0%) - BMPR2 n = 3 - ENG n = 1 - CPS1 n = 1 - ABCC8/SMAD1 n = 1 - SOX17 n = 1	8.0

## Data Availability

Not applicable.

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
