# Peer review of "Clinical Implications of the Genetic Background in Pediatric Pulmonary Arterial Hypertension: Data from the Spanish REHIPED Registry"

_ijms, 2022, doi:10.3390/ijms231810433_

Round 1

Reviewer 1 Report

Cruz-Utrilla et al. conducted extensive genetic screening on pediatric patients with pulmonary arterial hypertension (PAH) and analyzed the impact of the conducted testing on clinical classification and prognosis in this cohort.

Major comments:

In your study, you nicely delineate the effect of genetic testing on the classification and prognosis of pediatric PAH patients. However, it remains unclear whether patients benefit from being re-assigned to the new group in the terms of a more appropriate therapy algorithm, management of symptoms, and consequently quality of life and survival. Would it be possible to compare the survival of your patients classified based on genetic testing to the predicted survival for the group where they were included based on the clinical presentation? For example, after you divided the IPAH group into several different groups (IPAH, HPAH, TBX4...), did these patients on average do better than they would if they stayed classified in the IPAH group?

If it is not possible to make such a comparison, please comment on whether this can be done in the future and discuss the issue in the text. Without the improvement in the disease treatment, the clinical impact of the study seems limited to the better prediction of the inevitable outcome.

Minor comments:

All abbreviations should be explained (spelled out) the first time they appear in the text.

Author Response

Cruz-Utrilla et al. conducted extensive genetic screening on pediatric patients with pulmonary arterial hypertension (PAH) and analyzed the impact of the conducted testing on clinical classification and prognosis in this cohort.

Major comments:

In your study, you nicely delineate the effect of genetic testing on the classification and prognosis of pediatric PAH patients. However, it remains unclear whether patients benefit from being re-assigned to the new group in the terms of a more appropriate therapy algorithm, management of symptoms, and consequently quality of life and survival. Would it be possible to compare the survival of your patients classified based on genetic testing to the predicted survival for the group where they were included based on the clinical presentation? For example, after you divided the IPAH group into several different groups (IPAH, HPAH, TBX4...), did these patients on average do better than they would if they stayed classified in the IPAH group?

If it is not possible to make such a comparison, please comment on whether this can be done in the future and discuss the issue in the text. Without the improvement in the disease treatment, the clinical impact of the study seems limited to the better prediction of the inevitable outcome.

We thank the reviewer for this interesting question. As reflected in Figure 4, and also in the section 3.5. “Differences in survival free of death or lung transplantation according to the clinical classification and genetics after genetic testing”, we found global differences in survival, suggesting a lower survival in patients classified as PVOD or PH associated to MSD after genetic analysis (Figure 4A). We also tried to analyze differences depending on the specific gene found (Figure 4B) (lines 335-346). Although there are global differences and some genes seems to be related with unfavorable outcomes, unfortunately the number of patients in each group make difficult to interpret these results.

Pulmonary arterial hypertension is a hemodynamic condition in which the increase in pulmonary pressure can be produced by different pathogenic pathways and mechanisms. Nevertheless, at baseline, many of them are classified as “idiopathic PAH”. The genetic testing is allowing us to discriminate different PH forms, and if there are substantial differences in their management. For instance, pulmonary vasodilators, the usual treatment for idiopathic PAH, can produce fatal pulmonary oedema in PVOD patients, and in some multisystemic disorders, as in patients with NFU1 pathogenic variants; Patients with PH and cobalamin deficiency can resolve their PH with the specific treatment for the condition, avoiding lung transplantation; In patients with congenital heart diseases, the knowledge of the genetic background can prevent us from the closure of the defect, that could be deleterious; In neonates or infants with “idiopathic PAH” the finding of FOXF1 variants (alveolocapillary dysplasia) can accelerate either lung transplantation listing or redirection of care. Unfortunately, our study cannot prove that the genetic testing improves outcomes, as it was not designed with this purpose. We will need several years and the generalization of genetic testing to see an improvement in classification and management of patients based on molecular diagnosis.

We have added a comment on future perspectives of genetic testing for clinical reclassification and prognosis (lines 426-430 and 497-502).

Minor comments:

All abbreviations should be explained (spelled out) the first time they appear in the text.

It has been corrected. We have also included all abbreviations at the beginning (lines 73-80).

Reviewer 2 Report

Comments to the authors

Page 1 line 47: the word consecutive in this context signifies that all patients submitted to REHIPED had a similar chance to be  enrolled in your study, or are there still many possible biases, for instance leaving out the  many of the most severe cases, because they might have died shorly after diagnosis, and  died shortly after having been entered into REHIPED

Page2 line 55: 44 patients(44.9%) refer to table 2, which in my opinion should be moved to the supplementary information.

Page 3 line 103 : explain abbreviations the first time or add a list of  all abbreviations used throughout the paper at the beginning. DLD= developmental Lung Disease

Page 3 line 110 GBE1:  I  could not find a reference of GSD4 or polyglucosan body disease with pulmonary hypertension, so this  association is probably new; PAH has been described in GSD2 and GSD1 lending some credibility to the association you imply. Maybe a remark or reference would be in place here .  MECP2 duplications in affected males with PAH have been described on more than one occasion, but I am unfamiliar with classical  LoF MECP2 mutations in  females with Rett syndrome please  provide proper refrerence, MMACHC ( multiple publications, please provide ref), NFU1 ( multiple publications please provide proper reference).  VHL , please provide reference and state whether the  VHL mutation is biallelic, as has been described earlier.

Page 3 fig1. Very small black letters in filter 1 on a dark grey background not legible in a print version

Page 4 fig 2 The light grey box with the data on the 8  non-valid patients remains barely legible even at quite extreme magnification levels I believe it reads:    group 3PH (n=6)

                                                                                                                              Group 2 ( pulmonary vein stenosis)

                                                                                                                              (n=2) ATP13A3 vus

                                                                                                                                        NOTCHt1 vus

Even if I did decipher this correctly, however interesting I find the vusses, I do not understand what the remarks exactly mean, and I still have no clue, why these 8 patients were regarded as non-valid for enrolment in the study.

Page 4 line 170: strange wording please change majoritarian to predominant

Page 5 table 1.): very poorly aligned, therefore   some numbers that should add –up , do not!

In-proper use of the word race, please change to origin ,descendence extraction or population. Why are arabs mentioned  in the table at all, when there are no patients of Arab origin in the cohort.

 I do not intuitively understand the p-values, as I do not immediately understand what is compared to what, please clarify. I understand the median onset age is significantly different in different in different sub groups(p<0.001), does this hold true for all categories or just for PH-MSD when compared to the entire cohort? In In the ’race’  section I take it that significance refers to the fact that Caucasians  are overrepresented in the entire cohort? The fact that the roma group just 8,2% of the entire cohort  explains 75% of the PVOD group, did not reach significance?  the  In mode of ascertainment/diagnosis p=0,001 means that PVOD was relatively often diagnosed through an affected family member?

Page 6 line 201: Were you able to perform CNV calling on your exome data? Is it possible that the IPAH patient with a  single pahogecic EIF2AK4 variant may have had a partial deletion on the other allele?

Page 6 line 206 I believe that AQP1 as a bonafide PAH gene still needs a reference.

Page 6 line 210-13  I wondered where the HHT patients are in figure 2, I presume under MSD?

Page 7 line 220-222:       When posiible add references  to substantiate the role of the 3 genes mentioned in PAH.

Page7 line 248-249:         Provide references  for the role of the genes mentioned in PAH.

Page 8:  figure 4. Absense of color in figure 4 makes it illegible

Page 8 table 2. However interesting could be moved in my opinion to the supplementary data section

Page 10 line 305:             The Roma or Romani are not a race but an imbred nomadic population  or ethnic group

 There are some quite annoying inconsistencies f.e. figure 2 they have 98 valid patient that they subdivide in  5 categories, adding those up they only have 97 patients, so they already lost one patient before even starting to reclassify them based on the genetics. Subsequently the 51  suspected IPAH patients after genetics are  classified into in 6  categories, adding up to 52 patient, so they gain one lost patient here. My point being that if you start with 98 patients, they can not become 97 when dividing them over 5 clinical diagnostic categories, and subsequently become 98 again after genetic analysis. So this figure I am afraid has to be redone.

 As I understand it all genetically tested patients did either display a P or LP variant on the genepanels, which expanded over time, or had a singleton open exome. In that last step they identified pathogenetic variants in genetic conditions that are not usually associated with PAH like several metabolic disorders and in assoviation with MECP2 variants ( because I do not have acces to the supplementary table E1. I can not judge whether or not these MECP2 variants are LOF variants in females with Rett syndrome, which I believe would be an absolute novelty and should be mentioned as such, or in males with a MECP2 duplication, which has been reported a few times, but still would  need a reference in my opinion.

We are still stuck in the genepanel stage  for clinical PAH genetics..

But,  while performing many open exomes for different conditions Mainly ID and multiple congenital abnormalities in our experience  singleton open exomes leave us with many likely pathogenic variants in genes of unknown function/significance (genes that are with the current knowledge are not yet disease genes). That’s why, to optimize interpretation, we still do a lot of trio analyses in our clinical exomes. The authors do not elaborate on what they encounter in their open exome step, that cannot easily be attributed to either PAH or any other known genetic disease, they merely state that they find variants pointing in the direction of more systemic conditions  that might sometimes be associated with PAH. Having done this study I feel that the authors should at least come with a recommendation of how to genetically tackle pediatric PAH,! Should we  always open the exomes if our panels do not solve the case in the clinic or do they propose to add rare metabolic diseases and syndromes (MECP2, VHL, GBE1 and MMACHC) to our PAH  gene panels? Therefore the authors should formulate some recommendation should every child with PAH have a singleton  open  clinical exome, trio open exome, or is it sufficient for now  to perform something like the HAPv3 gene panel. In order to make a statement about this, it would be nice if the authors added some  easy accessible information on the extra yield of open exome when the most extensive HAPv3 panel had come back negative. A think adding this would be extremely relevant extra information.

Author Response

Page 1 line 47: the word consecutive in this context signifies that all patients submitted to REHIPED had a similar chance to be enrolled in your study, or are there still many possible biases, for instance leaving out the many of the most severe cases, because they might have died shorly after diagnosis, and died shortly after having been entered into REHIPED

As suggested, we do not include the word consecutive in the new version of the manuscript. Although we try to include all consecutive patients in this registry, some patients could have not been included in this period of time as a consequence of being unstable or died shortly after diagnosis. On the contrary, more unstable patients could have been included in referral centres. This statement has also been included in limitations (lines 480-481).

Page2 line 55: 44 patients (44.9%) refer to table 2, which in my opinion should be moved to the supplementary information.

This work compares the results of genetic testing in the Spanish Pediatric Population with other similar cohorts worldwide. From a clinical point of view, we think is important to show the global diagnostic yield of genetic analyses in our cohort and compare them with some of the most relevant previous works. Table 2 could be an interesting and quick guide for other clinicians to compare our results with the results in other similar papers, and therefore we would like to maintain it in the main text, if possible.

Page 3 line 103: explain abbreviations the first time or add a list of  all abbreviations used throughout the paper at the beginning. DLD= developmental Lung Disease

It has been corrected. We have also included all abbreviations at the beginning (lines 73-80).

Page 3 line 110 GBE1:  I could not find a reference of GSD4 or polyglucosan body disease with pulmonary hypertension, so this association is probably new; PAH has been described in GSD2 and GSD1 lending some credibility to the association you imply. Maybe a remark or reference would be in place here. MECP2 duplications in affected males with PAH have been described on more than one occasion, but I am unfamiliar with classical LoF MECP2 mutations in females with Rett syndrome please  provide proper reference, MMACHC (multiple publications, please provide ref), NFU1 ( multiple publications please provide proper reference).  VHL, please provide reference and state whether the VHL mutation is biallelic, as has been described earlier.

Thank you very much the reviewer for these interesting appreciations:

  • We have remarked in this new version of the manuscript the fact that this is the first time that a GBE1 variant is described in a case of neonatal PH pathologically confirmed, consistent with the diagnosis of a glycogen storage disease type IV (lines 433-438).
  • We have added a reference regarding MECP2 duplication syndrome and its association with PAH. Regarding LoF variants in MECP2, some studies found that lung samples of affected patients had multiple lesions that could be linked with the development of PH. Equally, rat models of Rett Syndrome demonstrated higher pulmonary vascular resistance and right ventricle hypertrophy in an unpublished PhD thesis (https://theses.gla.ac.uk/4097/1/2012Brockettphd.pdf). We have also added some evidence in this regard and have clarified the importance of our finding (lines 438-445).
  • We have actualized the references about MMACHC and NFU1 (references 30 and 32)
  • We have added recent evidence describing several cases of PAH patients with homozygous variants in VHL and compound heterozygous patients. In our case, the genetic study demonstrated a homozygous pathogenic variant in VHL instead of a heterozygous variant, as we wrote in the previous version. We have corrected the mistake in the genetic variant in the new version of the manuscript, both in the main text (lines 445-448) and in the Supplementary Table 1.

Page 3 fig1. Very small black letters in filter 1 on a dark grey background not legible in a print version

Figure 1 has been changed in the new version of the manuscript, as well as figure legend.

Page 4 fig 2 The light grey box with the data on the 8 non-valid patients remains barely legible even at quite extreme magnification levels I believe it reads:    group 3PH (n=6), Group 2 ( pulmonary vein stenosis), (n=2) ATP13A3 (vus), NOTCHt1 (vus). Even if I did decipher this correctly, however interesting I find the vusses, I do not understand what the remarks exactly mean, and I still have no clue, why these 8 patients were regarded as non-valid for enrolment in the study.

We have amplified the text in the box of excluded patients, and we have clarified the reasons for exclusion in the legend. Although these 8 cases had a genetic analysis done, we decided to exclude patients with a clinical diagnosis of group 2 PH (pulmonary vein stenoses) or group 3 PH (bronchopulmonary dysplasia as the main clinical feature), as the pathogenic mechanisms for pulmonary hypertension in group 2 are mainly related to elevated pulmonare wedge pressures and not to genetic predisposition. Similarly, patients with bronchopulmonary dysplasia usually present PH with a distinct pathogenic mechanism, related to extreme, and a distinct temporal course, improving or resolving with patient growth. Additionally, in those last patients no genetic background has been identified so far.

Page 4 line 170: strange wording please change majoritarian to predominant

Done.

Page 5 table 1.): very poorly aligned, therefore some numbers that should add –up , do not!

We have corrected the table 1.

In-proper use of the word race, please change to origin ,descendence extraction or population. Why are arabs mentioned  in the table at all, when there are no patients of Arab origin in the cohort.

We have changed race in all the manuscript, using the proposed words. In the whole cohort there were two cases of Arabic origin in the group of PH associated to multisystemic disorders.

 I do not intuitively understand the p-values, as I do not immediately understand what is compared to what, please clarify. I understand the median onset age is significantly different in different in different sub groups(p<0.001), does this hold true for all categories or just for PH-MSD when compared to the entire cohort? In In the ’race’  section I take it that significance refers to the fact that Caucasians  are overrepresented in the entire cohort? The fact that the roma group just 8,2% of the entire cohort  explains 75% of the PVOD group, did not reach significance?  the  In mode of ascertainment/diagnosis p=0,001 means that PVOD was relatively often diagnosed through an affected family member?

In the “Descendance” section, p-value refers to the fact that there are global differences between clinical PH groups, and do not point out where the differences are, or if Roma group is more frequent in the PVOD group. In the case of Mode of Diagnosis, there are also global differences between groups, suggesting that PVOD and heritable PAH are diagnosed more frequently by family screening, and are not so frequently the index case. We have clarified this in the table legend and in the Results section (lines 217-223). 

Page 6 line 201: Were you able to perform CNV calling on your exome data? Is it possible that the IPAH patient with a single pahogecic EIF2AK4 variant may have had a partial deletion on the other allele?

Yes, in all patients the study of CNVs has been carried out and, specifically in this patient, we have not detected any. In addition, in patients such as the case of this patient in which we only detected a heterozygous variant in EIF2AK4, we performed an exhaustive review of the BAM file to rule out the presence of variants that had been discarded due to low quality in the filtering algorithm designed. However, we did not detect any second variant that could explain a compound heterozygosity.

Page 6 line 206 I believe that AQP1 as a bonafide PAH gene still needs a reference.

We have included a reference of this specific family with an AQP1 gene variant and review of the literature (line 262, reference 16).

Gallego-Zazo N, Cruz-Utrilla A, Del Cerro MJ, et al. Description of Two New Cases of AQP1 Related Pulmonary Arterial Hypertension and Review of the Literature. Genes (Basel). 2022;13(5). doi:10.3390/genes13050927

Page 6 line 210-13 I wondered where the HHT patients are in figure 2, I presume under MSD?

We found 2 cases of HHT. Both were initially classified as idiopathic PAH and reclassified after genetic testing as heritable PAH forms. We have clarified this in the new version of the manuscript (line 268).

Page 7 line 220-222: When posiible add references to substantiate the role of the 3 genes mentioned in PAH.

We included several references all over the manuscript regarding the mentioned genes.

Page7 line 248-249: Provide references for the role of the genes mentioned in PAH.

We included several references all over the manuscript regarding the mentioned genes.

Page 8:  figure 4. Absense of color in figure 4 makes it illegible

We have changed the figure, showing the original light-blue color.

Page 8 table 2. However interesting could be moved in my opinion to the supplementary data section

As referred in previous comments, the intention of the manuscript is to combine clinical and genetical data in Pediatric pulmonary hypertension. As a result, we think that a table comparing different manuscripts could be of special value for clinicians to compare these data in different populations. We thank the consideration of the reviewer that, if possible, we would like to maintain the table in the main text.

Page 10 line 305: The Roma or Romani are not a race but an imbred nomadic population or ethnic group

We have changed the term “race” across the work, using ethnicity instead of race

There are some quite annoying inconsistencies f.e. figure 2 they have 98 valid patient that they subdivide in 5 categories, adding those up they only have 97 patients, so they already lost one patient before even starting to reclassify them based on the genetics. Subsequently the 51 suspected IPAH patients after genetics are classified into in 6  categories, adding up to 52 patient, so they gain one lost patient here. My point being that if you start with 98 patients, they cannot become 97 when dividing them over 5 clinical diagnostic categories, and subsequently become 98 again after genetic analysis. So this figure I am afraid has to be redone.

The figure has been changed. The group of patients initially classified clinically as idiopathic PAH was composed of 52 patients instead of 51.

 As I understand it all genetically tested patients did either display a P or LP variant on the genepanels, which expanded over time, or had a singleton open exome. In that last step they identified pathogenetic variants in genetic conditions that are not usually associated with PAH like several metabolic disorders and in assoviation with MECP2 variants ( because I do not have acces to the supplementary table E1. I can not judge whether or not these MECP2 variants are LOF variants in females with Rett syndrome, which I believe would be an absolute novelty and should be mentioned as such, or in males with a MECP2 duplication, which has been reported a few times, but still would  need a reference in my opinion.

We thank again the reviewer for the suggestion. In this work we describe two female patients with LoF variants in MECP2, which is a new finding in the literature. As a result, we have included this suggestion in the Discussion (lines 438-445).

We are still stuck in the genepanel stage for clinical PAH genetics..

But, while performing many open exomes for different conditions Mainly ID and multiple congenital abnormalities in our experience  singleton open exomes leave us with many likely pathogenic variants in genes of unknown function/significance (genes that are with the current knowledge are not yet disease genes). That’s why, to optimize interpretation, we still do a lot of trio analyses in our clinical exomes. The authors do not elaborate on what they encounter in their open exome step, that cannot easily be attributed to either PAH or any other known genetic disease, they merely state that they find variants pointing in the direction of more systemic conditions  that might sometimes be associated with PAH. Having done this study I feel that the authors should at least come with a recommendation of how to genetically tackle pediatric PAH,! Should we  always open the exomes if our panels do not solve the case in the clinic or do they propose to add rare metabolic diseases and syndromes (MECP2, VHL, GBE1 and MMACHC) to our PAH  gene panels? Therefore the authors should formulate some recommendation should every child with PAH have a singleton  open  clinical exome, trio open exome, or is it sufficient for now  to perform something like the HAPv3 gene panel. In order to make a statement about this, it would be nice if the authors added some easy accessible information on the extra yield of open exome when the most extensive HAPv3 panel had come back negative. A think adding this would be extremely relevant extra information.

We have included a sentence in the discussion section to address the interesting question suggested to us by the reviewer (lines 468-476). From our experience, we suggest the following approach for the genetic diagnosis of pediatric cases. Ideally, the genetic testing should start with the index case and, when possible, study the patient with a gene panel which include BMPR2 and other genes with high evidence of association with the development of PAH. Depending on available resources, for cases without an identified variant after the massive paralleled sequencing, it would be advisable exome or genome sequencing. If it is possible, the approach should include parents in order to address the identification of de novo variants. When Whole Exome Sequencing (WES) is performed, it is easier to review patient results with additional new genes identified. In addition, the study through WES is useful in the research scenario, in the search for new genes potentially involved in the disease, as we have propose in previous publications.

Round 2

Reviewer 1 Report

No further comments